# "Special Relativity" of Image Aesthetics Assessment: a Preliminary Empirical Perspective

## ABSTRACT

Image aesthetics assessment (IAA) primarily examines image quality from a user-centric perspective and can be applied to guide various applications, including image capture, recommendation, and enhancement. The fundamental issue in IAA revolves around the quantification of image aesthetics. Existing methodologies rely on assigning a scalar (or a distribution) to represent aesthetic value based on conventional practices, which confines this scalar within a specific range and artificially labels it. However, conventional methods rarely incorporate research on interpretability, particularly lacking systematic responses to the following three fundamental questions: 1) Can aesthetic qualities be quantified? 2) What is the nature of quantifying aesthetics? 3) How can aesthetics be accurately quantified? In this paper, we present a law called "Special Relativity" of IAA (SR-IAA) that addresses the aforementioned core questions. We have developed a **M**ulti-**A**ttribute **I**AA Framework (MAINet), which serves as a preliminary validation for SR-IAA within the existing datasets and achieves state-of-the-art (SOTA) performance. Specifically, our metrics on multi-attribute assessment outperform the second-best performance by 8.06% (AADB), 1.67% (PARA), and 2.44% (SPAQ) in terms of SRCC. We anticipate that our research will offer innovative theoretical guidance to the IAA research community. Codes are available in **the supplementary material**.

## KEYWORDS

image aesthetics assessment; neural networks

## 1 INTRODUCTION

Image aesthetics assessment (IAA) is a fundamental task in the field of computer vision. IAA primarily examines the quality of images from a user-centric perspective, rather than primarily focusing on signal distortion as conducted by image quality assessment (IQA).

In recent years, the surge in social media usage and advancements in digital photography have led to an increased demand for IAA, aiming to automatically evaluate whether an image aligns with users' aesthetic preferences [1–3]. In fact, IAA can be applied to guide various applications, including image capture (e.g., enhancing the photographic capabilities of smartphones and cameras [4–6]), recommendation (e.g., recommending social medias or advertising), and enhancement (e.g., facilitating the creation of visually captivating images [7, 8] with the rapid advancement of AI-generated content).

*ACM MM, 2024, Melbourne, Australia*
© 2024 Copyright held by the owner/author(s). Publication rights licensed to ACM.
ACM ISBN 978-x-xxxx-xxxx-x/YY/MM
https://doi.org/10.1145/nnnnnnn.nnnnnnn

The prevailing IAA methodologies still resort to assigning a scalar (or regressing a distribution to a scalar with maximum probability) to represent aesthetic value based on conventional practices, which confines this scalar within a specific range and artificially labels it. However, these methodologies are predicated on ***an implicit assumption*** that all images can be linearly ranked according to their aesthetic qualities; however, this assumption may be ***ill-posed*** itself.

To our knowledge, conventional methods rarely incorporate research on interpretability, particularly lacking systematic responses to the following three fundamental questions around the quantification of image aesthetics:

**I. Can aesthetic qualities be quantified?** Aesthetics is an abstract concept that lacks concreteness and is characterized by its inherent uncertainty: aesthetic perceptions vary among viewers and can even fluctuate for the same viewer over time; moreover, randomness is often considered when evaluating image aesthetics. Some researchers argue that the scientific method, known for its objectivity and rationality, is inadequate for quantifying aesthetics due to its subjective and uncertain nature [9]. Therefore, quantifying aesthetics appears to be impractical without establishing certainty.

**II. What is the nature of quantifying aesthetics?** The prevailing methods of quantification involve scoring images, which serves two essential purposes: positioning a sample within the entire sample space and assessing aesthetic differences between samples. Existing methods quantify aesthetics based on ***numerical scores or distribution*** and their differences, overlooking the non-transitive nature of IAA (confused by the transitivity of mathematical inequalities). Specifically, if image A is considered more aesthetically pleasing than image B, and image B more so than image C, it does not necessarily imply that image A is more pleasing than image C.

**III. How can aesthetics be accurately quantified?** Current annotation methods are evidently inadequate for achieving this quantification because they conflate two essential purposes of quantification. Assigning a numerical score (e.g., ranging from 0 to 10) to an isolated sample is a considerable challenge, as it is difficult for viewers to determine an accurate score for both purposes of quantification. Furthermore, the act of confining quantification within a specific range is inherently illogical, as it can be easily disproven: we can always encounter an image that surpasses all the samples in the dataset in terms of either ugliness or beauty.

This paper aims to offer comprehensive and systematic responses to the aforementioned three fundamental issues concerning aesthetic quantification, while also presenting conclusive findings and viable solutions. However, it is important to acknowledge that due to the limitations inherent in existing datasets and the length constraint of a conference paper, our verification can only be considered partial and preliminary in nature. Our contributions can be summarized as follows:

- To the best of our knowledge, this study represents the first comprehensive and systematic analysis addressing three fundamental issues in IAA.
- This paper proposes a law called "Special Relativity" of IAA (SR-IAA), which investigates three fundamental issues around quantifying aesthetics. The term "special" implies that the law is based on a framework constructed by aesthetic qualities, viewers, cameras, and photographers, but does not account for temporal (e.g. across eras or ages) or spacial (environmental) influences, nor dynamic alterations in human neuronal structures and their synaptic connections.
- Guided by SR-IAA, we have developed a **M**ulti-**A**ttribute **I**AA Framework (MAINet), which serves as a preliminary validation for SR-IAA within the existing datasets and achieves SOTA performance.

## 2 RELATED WORKS

### 2.1 Quantification of Aesthetics

**Possibility.** The subjective nature of image aesthetics poses the primary challenge in IAA, as it can be influenced by cultural background, personal experiences, education attainment and psychological state [10]. Some researchers argue that the scientific methods may not be sufficient to quantify aesthetics due to its subjectivity and uncertainty [9]. Conversely, the presence of the "Law of Large Numbers" in statistics supports the feasibility of using objective computational methods. Additionally, some studies aim to integrate more objective criteria into IAA by considering attributes such as semantic content and artistic aspects, with the goal of providing a reliable basis for evaluating aesthetic properties of different photographs [11, 12].

According to the aforementioned viewpoints, the crux of the dispute over aesthetic quantification lies in determining if reliability can be derived amidst uncertainties. Existing methodologies mainly rely on statistical perspectives like "Law of Large Numbers" or "majority consensus". However, this kind of statistical certainty is not only significantly constrained by the number and representativeness of the sampled population, but also it lacks effectiveness in addressing personalized individuals.

**Methodologies.** In current research, general IAA tasks are typically classified into three categories: binary classification (positive or negative aesthetics) [13, 14], aesthetic score regression [15, 16] and score distribution prediction [17–19]. On the other hand, personalized IAA customizes an aesthetic model to suit individual user preferences [20, 21]. Correspondingly, existing datasets [22–25] aim to collect more comprehensive scores by increasing the number of annotators or broaden the range of aesthetic attributes. Moreover, other studies [26, 27] have harnessed self-supervised learning paradigms to formulate image sequences exhibiting subtle aesthetic fluctuations for data augmentation.

However, it is often overlooked that the prevailing methods of quantification involve scoring images, serving two essential purposes: positioning samples within the entire sample space and assessing aesthetic differences between them. Clearly, existing methods struggle to effectively balance these two purposes using a single scalar, often resulting in neglecting one or the other.

**Inspirations.** Comparison-based methods [28–32] are useful to learn metrics for quantifying perceptual concepts, like urban appearance quantification [33]. In the past decade, comparison-based approaches have proven effective in the IAA field. Kong *et al.* [22] introduced a loss function called "pairwise ranking loss" to ensure accurate image ranking. Lee *et al.* [34] developed a unified approach for scoring regression, binary classification, and personalization based on comparison. Ko *et al.* [35] proposed using pairwise comparison to evaluate image aesthetics.

However, the existing comparison-based methods, whether tailored for IAA [22, 34, 35] or more generally applicable in other fields [28–32], although capable of providing relative quantitative metrics and circumventing the fixed range limitation of absolute quantitative metrics, presuppose an ordered data structure without considering the unique characteristic of "aesthetic inequality does not satisfy transitivity". This paper endeavors to address this issue through specialized processing of existing aesthetic datasets.

Furthermore, drawing inspiration from Shin *et al.*[28], who introduced the concept of "using two reference instances to form a window," and Burges *et al.*[29], who posited that "the relative order of items is more significant than their absolute scores," we developed two strategies to effectively overcome the limitations imposed by reliance on absolute scores and have clearly delineated the purposes of aesthetic quantification. Firstly, we ***adaptively select reference images*** to position a sample within the entire sample space. Secondly, we assess aesthetic differences between samples by ***perceiving differences in aesthetic attributes***.

### 2.2 Multi-attribute IAA

Multi-attribute IAA takes into account various attributes or features in an image, such as color, contrast, composition, etc. It incorporates the photographic rules, camera parameters, and viewer preferences more comprehensively and provides multi-attribute explanations. Early works [14, 36] utilized hand-crafted features based on standard photographic rules, such as the rule of thirds, colorfulness, or saturation, to distinguish between aesthetically pleasing and displeasing images. Recently, the availability of multi-attribute datasets [22–24] allow multi-column networks [37] and multi-task universal models [22, 38, 39] to evaluate various aesthetic attributes. Notably, Transformer or CLIP models [1, 40–44] have been widely applied to extract aesthetic information and map visual features to annotated labels. Multi-attribute IAA primarily focuses on measuring aesthetic differences rather than quantifying individual sample positions.

However, aesthetic attributes are typically delineated based on individuals' perceptual knowledge, and comprehensively identifying all aesthetic attributes within the existing cognitive framework is essentially unattainable. Furthermore, the definition and selection of aesthetic attributes depend on distinct aesthetic values, thereby resulting in existing aesthetic datasets with inherent limitations in scalability and consistency. This significantly hinders any modifications or additions to aesthetic attributes, necessitating a comprehensive reworking of the datasets.

To address the challenges of multi-attribute IAA while validating our law, we propose MAINet for more accurate generation of aesthetic scores. Additionally, guided by our law, the issue of modifying or adding aesthetic attributes is resolved in an unsupervised manner.

## 3 LAW CALLED SR-IAA

Regarding a rational viewer:

*I. Image aesthetics can be quantified by consistently and definitively determining the relative preference between two comparable images within a given duration. However, aesthetic inequalities do not adhere to the transitivity observed in mathematical inequalities.*

Note: The duration varies among individuals depending on the extent to which their neurons are adequately stimulated to significantly influence their aesthetic preference; If the duration is brief without additional stimulus input, it indicates similar aesthetic in both images. Furthermore, the aesthetic comparability of any two images cannot be guaranteed, as evidenced by the contrasting themes of probability statistics and natural scenery.

*II. Quantifying image aesthetics primarily aims to simulate two abilities: perceiving aesthetics without any reference and perceiving aesthetics with a reference.*

Note: The first ability involves "locating" the input sample within an "experience sample" space, while the second ability involves calculating aesthetic differences between two input samples.

*III. Quantifying aesthetic perception without any reference can be represented by establishing pairwise relative relationships among N samples, while the quantification with a reference involves identifying variations in multiple aesthetic attributes between two samples.*

Note: The value of $N$ should be large enough, with an aesthetically uniformly distributed sample set; the aesthetic attributes, which are defined and selected based on individual aesthetic values, can all be assigned to three roles [45] : camera, photographer, and viewer.

Due to length constraints, we provide an outline of foundations underlying the law (hypothesis) here: 1) *philosophically*, SR-IAA is the only comprehensive sampling approach for assessing image aesthetics (addressing both goals of aesthetic quantification), unlike others (achieving at most one goal of aesthetic quantification), to our knowledge; 2) *psychologically*, the biological instinct of "stimulus response" and "seeking advantage and avoiding harm" manifests in psychology as the cognitive process of "pursuing beauty while avoiding ugliness"; 3) *mathematically*, aesthetic inequalities fail to satisfy the transitivity property; hence, traditional quantification by scoring is inherently inaccurate, while SR-IAA circumvents this issue. For further elucidation, please consult the Appendix A.1. It is conceivable that another future publication may be necessary to comprehensively expound and substantiate the underpinnings of SR-IAA; however, *within the scope of this paper, proposing the law (hypotheses) suffices without compromising its integrity.*

Based on this premise, we posit that SR-IAA would be well-founded, we propose a framework for multi-attribute IAA and aim to preliminarily validate the law by using it as guidance to enhance IAA performance on the existing datasets.

## 4 MAINET FOR VALIDATION

### 4.1 Limitations and Compromises

The available datasets are primarily constrained by the following three limitations, necessitating tailored compromises to be made, which is why we refer to it as a "Preliminary Empirical Perspective."

- **Limitation I:** Current data annotation systems linely rank images based on their scores which contradicts "SR-IAA I."
**Compromise I:** Fortunately, we have observed that image sequences with a common theme exhibit a certain level of transitivity like mathematical inequalities, which is the fundamental reason behind the effectiveness of theme-based TANet [1]. Therefore, guided by "SR-IAA I & II," the aesthetic quality of an image can be approximated by comparing it to other ones with the same theme that are more or less pleasing.
- **Limitation II:** Existing datasets exhibit limited scalability in terms of new or redefined multi-attributes.
**Compromise II:** The aesthetic difference between images, as stated in "SR-IAA III," is determined by multiple attributes. To measure these attributes, the idea of "Compromise I" can also be employed. When extending to a new or redefined attribute, quantifying the editing degree of an image can approximate the difference in aesthetic attributes between the original and edited versions. This approach enables unsupervised scalability of aesthetic attributes.
- **Limitation III:** The majority of available datasets are annotated using average ratings from multiple assessors.
**Compromise III:** The average rating can be approximated to that of a "being smoothed" viewer, although this approximation may present new challenges (refer to Appendix A.1.III).

### 4.2 Problem Definition

For any given input image $x$, MAINet necessitates the identification of a lower reference image ($R_l$) and an upper reference image ($R_u$) within a common theme in order to establish aesthetic boundaries. Subsequently, we conduct a regression task to evaluate aesthetic differences within a defined triplet $\{x, R_l, R_u\}$.

$$\mathcal{L}_{diff}(S, y) = \min_{\theta} |y - \mathcal{M}(S; \theta)|, \qquad (1)$$

where the set $S = \{x, R_l, R_r\}$ represents the triplet, $y$ is the ground truth, and $\theta$ is the parameter to be optimized for model $\mathcal{M}$.

### 4.3 Architecture of MAINet

The architecture is illustrated in Fig.1. Given an RGB image $x \in \mathbb{R}^{H \times W \times C}$, the Reference Selection Module (RSM) selects two reference images within a common theme, forming a triplet denoted as $S \in \mathbb{R}^{3 \times H \times W \times C}$. Subsequently, the images in $S$ are split into patches of size $\frac{H}{32} \times \frac{W}{32}$ using a patchify strategy. These patches are then processed by three modules: the Position Identification Module (PIM), which assigns each patch a position through a learnable matrix $P \in \mathbb{R}^{3 \times \frac{H}{32} \times \frac{W}{32}}$; the Patch Embedding Module (PEM), which converts these patches into embeddings; and finally, the Attribute Perception Module (APM), which analyzes their attributes for comparison purposes. Ultimately, these module outputs are integrated and fed into a decoder to predict differences in aesthetic quality between images before being converted into an aesthetic score.

*4.3.1 Reference Selection Module (RSM)* . The task assigned to RSM, in accordance with **Compromis I**, is to select two reference images that share a common theme with an input image. To simplify this process, RSM has been designed to progressively master four-level comparison tasks. The complexity of each task depends

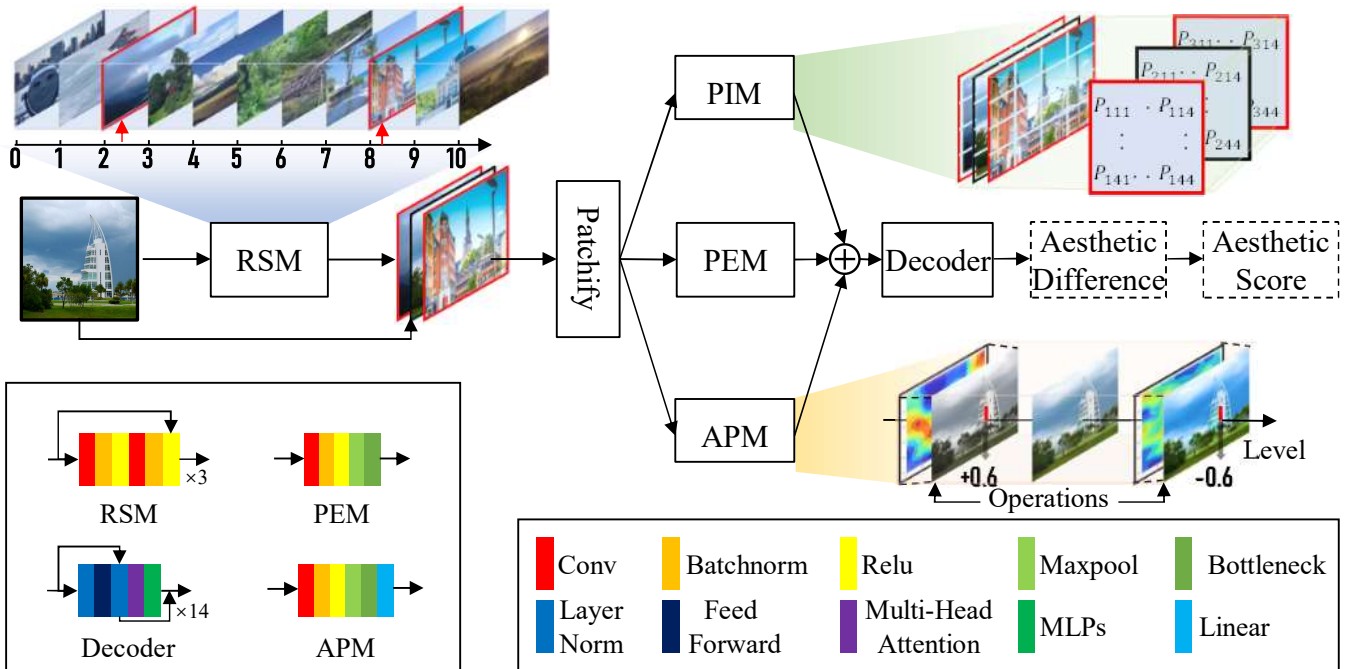

**Figure 1: Architecture of MAINet. It comprises 5 modules: RSM selects reference images, while PIM, PEM, and APM learn position and attribute information; the decoder combines features to predict aesthetic differences. Please refer to Sec. 4.3 for more details.**

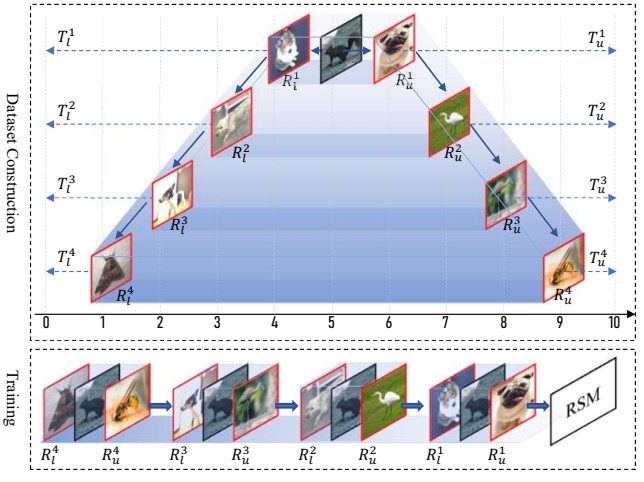

**Figure 2: Dataset construction for RSM. The dataset was structured from difficult to easy instances, but the training methodology should start with simpler cases and progress to harder ones.**

on the aesthetic differences between the reference and input images. The simpler the task, the greater the aesthetic difference between the reference images and the input image. Conversely, finding references that possess similar appeal as the input image presents a more challenging task.

**I. Dataset Construction Strategy for RSM.**

The process of creating training samples for an input image $x$ is illustrated in Fig. 2 through a four-tier comparison. For each tier $e$, two semantically similar reference images $R_l^e$ and $R_u^e$ are selected, along with their respective scores determined by $T_l^e$ and $T_u^e$. These, combined with the input image $x$, form a training sample denoted as $S^e = (x, R_l^e, R_u^e, T_l^e, T_u^e)$, where $e \in \{1, 2, 3, 4\}$. The training samples are then organized into four datasets denoted as $D^e = \{S_i^e\}$, where the index range of $i$ is from 1 to N (the total number of input images) and the tiers span from 1 to 4. The detailed construction method for this dataset can be found in Algorithm 1.

It is essential to note that we ensured thematic consistency during the dataset construction phase. In selecting images, we utilized a pre-trained Theme Understanding Network [1] to filter and compile image combinations with similar themes as the final training data for the RSM. The considered themes span across 47 categories, including landscape, plant, and human. During the inference stage, we employ a consistent strategy to ensure that the reference images compared with the input images share the common theme.

**II. Training Strategy for RSM.**

In the training phase, RSM sequentially approaches tasks from the simplest ($e = 4$) to the most complex ($e = 1$) training datasets. For each difficulty level $e$, RSM processes a training sample $(x, T_l^e, T_u^e)$ and generates the output $(B_l^e, B_u^e)$. In the final level, the gap between $T_l^1$ and $T_u^1$ is 1. This process can be described by the following formulation:

$$\mathcal{L}_{loc} = \frac{1}{2}((T_l^e - B_l^e)^2 + (T_u^e - B_u^e)^2). \tag{2}$$

Finally, after sequential training across four tiers, the trained RSM is capable of predicting matched boundaries $(B_l^1, B_u^1)$ that closely

approximate $(T_l^1, T_u^1)$, indicating that the difference between $B_l^1$ and $B_u^1$ is also near 1. Subsequently, it selects corresponding reference images $(R_l^1, R_u^1)$ based on these boundaries $(B_l^1, B_u^1)$. Appendix A.2 elucidates the reliability and accuracy of triplet data, additionally providing illustrative comparisons.

By employing a four-tier training strategy, we innovatively endowed the model with the capability for self-correction. That is, although the model may err in the predictions at each tier, the likelihood of errors occurring across all four tiers is minimal. This significantly enhances the accuracy of the network's final prediction, thereby establishing certainty in aesthetic assessments. Moreover, selecting images of the common theme narrows the scope of the solution space to compare within the common thematic context.

---

**Algorithm 1** Constructing Triplets for RSM Training

---

**Input:** Dataset: $\{(x_i, y_i)\}, i \in \{1, 2, ..., N\}$
**Output:** $\{S_i^e\}, e \in \{1, 2, 3, 4\}, i \in \{1, 2, ..., N\}$
1: // $Get\_score(x)$: Get the score of image $x$
2: // $Select(x, a)$: Select an image that have common theme with $x$ and ensure its score is less than $a$
3: $i \leftarrow 1$, $e \leftarrow 1$
4: **while** $i \leq N$ **do**
5:     $x \leftarrow x_i$
6:     $a \leftarrow \lfloor y_i \rfloor$, $b \leftarrow \lceil y_i \rceil$
7:     **while** $e \leq 4$ **do**
8:        $R_l^e \leftarrow Select(x, a)$, $R_u^e \leftarrow Select(x, b)$
9:        $T_l^e \leftarrow Get\_score(R_l^e)$, $T_u^e \leftarrow Get\_score(R_u^e)$
10:       $S_i^e \leftarrow (x, R_l^e, R_u^e, T_l^e, T_u^e)$
11:       $e \leftarrow e + 1$, $a \leftarrow a - 1$, $b \leftarrow b + 1$
12:       **if** $a = 0$ **then**
13:          $a \leftarrow 1$, $b \leftarrow b + 1$
14:       **else if** $b = 11$ **then**
15:          $a \leftarrow a - 1$, $b \leftarrow 10$
16:       **end if**
17:     **end while**
18:     $i \leftarrow i + 1$
19: **end while**

---

*4.3.2 Attribute Perception Module (APM).* Aesthetic differences between two images are determined by aesthetic attributes (refer to "SR-IAA III"), which should be included in aesthetic comparisons [46]. According to **Compromis II**, APM uses a self-supervised training approach to predict editing operations and intensities, enhancing its comprehension of different attributes.

**I. Dataset Construction for APM.**

The latest research reveals that specific image editing operations have distinct effects on image attributes, influencing the aesthetics [26, 27]. Identifying the types and parameters of these editing operations applied to synthetic images can help models understand aesthetic features. With this goal in mind, we introduce certain tasks.

Fig. 3 demonstrates the utilization of 21 operations $O$ to identify the 7 attributes. These operations can be adjusted at 8 different intensity levels $L$, with cropping operations having only 2 intensity levels. The effectiveness of operations with different intensity levels can be consulted in Appendix A.4.

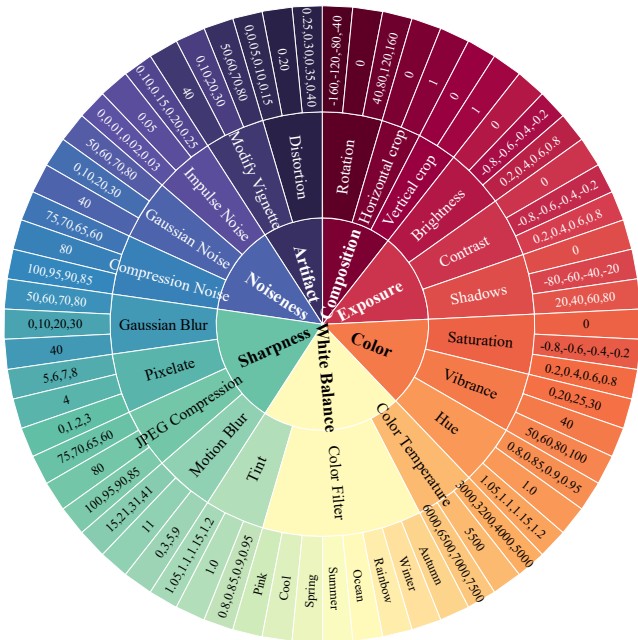

**Figure 3: The editing operations and intensities applied to images enable the generation of dataset for self-supervised training.**

It should be noted that some operation intensenty levels appear excessively strong, making them improbable in real photographs (for example, Gaussian noise levels of 50). However, it is essential to clarify that APM does not predict aesthetic score; instead, it solely predicts operation intensity. The employment of high-intensity parameters to ensure coverage of all possible extreme scenarios in real-world situations and to broaden the applicability of APM's feature perception is thoroughly justified.

Simultaneously, it is important to note that operation intensity does not directly correlate with aesthetic score, a higher level of operation intensity does not necessarily imply a superior aesthetic score. Therefore, operation intensity is not used as a direct basis for aesthetic comparison but rather serves as additional information for the decoder.

For dataset construction, a sample is defined as a triplet $(\tilde{x}, o, l)$, where $(o, l)$ indicates that the image $\tilde{x}$ was edited using operation $o \in O$ at parameter level $l \in L$. We refer to the dataset as the "augmented dataset" and validate its effectiveness in the ablation study section.

**II. Goal of Training .**

APM processes patches $p$ extracted from synthetic image $\tilde{x}$, and then predicts the probability distribution for the type and intensity level of each operation applied to these patches. We calculate the prediction losses using categorical cross-entropy:

$$\mathcal{L}_{attr} = -\left( \sum_{i=1}^{21} o_i \log(\hat{o}_i) + \sum_{i=1}^{8} l_i \log(\hat{l}_i) \right), \quad (3)$$

where $o$ and $l$ is respectively the ground-truth operation and intensities, while their predictions are represented by $\hat{o}$ and $\hat{l}$. The APM outputs an embedding of size $21 \times 8 \times D$.

*4.3.3 Patch Embedding Module (PEM).* After obtaining the reference images $R_l$, $R_u$ of the input image $x$, we divide the triplet $S \in \mathbb{R}^{3 \times H \times W \times C}$ into image patches of size $32 \times 32$. These patches are then encoded by PEM, with each patch linearly embedded by a 5-layer ResNet [47] (Fig.1).

*4.3.4 Position Identification Module (PIM).* In the classical transformer architecture, 2D-aware position embeddings are typically added to patch embeddings for preserving positional information. However, when dealing with triplets of images as input, it becomes easy to confuse patches from different images, potentially leading to comparison errors. To address this issue, we incorporate 3D patch positional information (PPI), enabling us to accurately determine the source image and corresponding position for each patch using a $3 \times D$-dimensional embedding.

Suppose the input resolution is $H \times W$, triplet $S$ will be split into $3 \times \frac{H}{32} \times \frac{W}{32}$ patches. We define PPI by a learnable matrix $P \in \mathbb{R}^{3 \times \frac{H}{32} \times \frac{W}{32}}$, for the patch at position i-th image of $S$, j-th row and k-th column, its position embedding is defined by the element at position $P_{i,j,k}$.

*4.3.5 Decoder of MAINet.* The decoder of MAINet processes embedding tokens to predict the aesthetic difference of the triplet.

**I. Structure of Decoder.** It receives various embedding tokens from PEM, PIM and APM, and consists of alternating layers of multi-head self-attention (MSA) and MLP blocks. LayerNorm (LN) is applied before every block, and residual connections after every block.

We append a [*class*] token at the beginning of the token sequence to capture important features from the entire sequence. Finally, we use MLPs with two layers and GELU non-linearity to predict image aesthetics:

$$\mathcal{M}((x, R_l, R_u)) = \text{LN}(\text{GELU}(\text{LN}(z_L^0))), \tag{4}$$

where the output of [*class*] token after the decoder, denoted as $z_L^0$, serves as the final output and is calculated as follows:

$$\begin{aligned}
\mathbf{z}_0 &= [class; \text{PEM}(p)] + \text{PIM}(p) + \text{APM}(p), \\
\mathbf{z}'_\ell &= \text{MSA}\left(\text{LN}\left(\mathbf{z}_{\ell-1}\right)\right) + \mathbf{z}_{\ell-1}, \ell = 1 \dots L, \\
\mathbf{z}_\ell &= \text{MLP}\left(\text{LN}\left(\mathbf{z}'_\ell\right)\right) + \mathbf{z}'_\ell, \ell = 1 \dots L.
\end{aligned} \tag{5}$$

**II. Dataset Reconstruction and Training Goal.**

The current datasets need to be reconstructed for the decoder to predict aesthetic differences within the existing data framework.

The given triplet sample $(x, R_l, R_u)$, $(y, y_l, y_u)$, consisting of input image $x$ and two reference images $R_l$ and $R_u$, along with their corresponding ground truth values $y, y_l, y_u$, is converted into an aesthetic difference value $y_t$ by Eq. (6).

$$y_t = \frac{y - y_l}{y_u - y_l} \quad (y_l < y < y_u). \tag{6}$$

However, if the dataset lacks manually labeled ground truth, we provide annotations based on the following assumption: any modifications made to high-quality images will degrade their original aesthetics. The extent of aesthetic degradation is relevant to the degree of image degradation. To create an "Original Triplet", we collect high-quality images and subject them to various synthetic operations and intensity levels.

For example, we rate the original image with high aesthetics as 10 which means without any degradation. When we apply two intensity levels of saturation operation (0.8 and 0.6), we produce two synthetic images with aesthetic levels of 2 ($= 10 \times (1 - 0.8)$) and 4 ($= 10 \times (1 - 0.6)$) respectively, in comparison to the original image. The values 2 and 4 are called as "aesthetic difference levels" rather than "aesthetic scores". This triplet can also be represented as $S_r = \{(x, R_l, R_u), (y, y_l, y_u)\}$ and can be transformed by Eq. (6). Finally, the training sample can be uniformly represented as $S_t = \{(x, R_l, R_u), y_t\}$.

During the training phase, both RSM and APM will be frozen; optimization will solely focus on PEM, PIM, and the decoder. The MAE loss is utilized to determine the difference in image aesthetics:

$$\mathcal{L}_{diff} = |y_t - \mathcal{M}((x, R_l, R_u))|. \tag{7}$$

Finally, in order to maintain consistency with other methods, we convert the aesthetic difference $y_t$ into an aesthetic score by applying Eq. (6) in reverse.

## 5 EXPERIMENTS

### 5.1 Settings

**Training Phases of MAINet.** MAINet consists of five learnable components: RSM, PIM, APM, PEM, and the decoder. To ensure stable training, we adopt a three-stage progressive approach. In the initial stage, our focus is on training the crucial RSM component for predicting matched boundaries and selecting reference images. Once this stage is completed, we freeze the parameters of RSM. Moving to the second stage involves self-supervised APM training to enhance attribute comprehension within the model. After completing this phase, we also freeze the parameters of APM. Finally, in the third stage of our approach, we train PIM, PEM and decoder components to predict aesthetic differences in images.

**Datasets.** We evaluate models on three datasets: AADB [22], SPAQ [23], and PARA [24]. The dataset has no special requirements in the first and third stages. For the dataset containing $N$ samples, we apply Algorithm 1 to generate $4N$ samples and split them into $0.8 \times 4N$ samples for training and $0.2 \times 4N$ samples for testing. In the second stage, we extract 600 images with aesthetic scores of 8.0 or higher from the AADB dataset. Following the methodology delineated in Sec. 4.3.2, we construct a dataset consisting of 93,600 samples. This dataset is subsequently split into training and testing sets, with 20% allocated for testing.

**Evaluation Metrics.** The performance is assessed using three popular evaluation metrics: Spearman rank correlation coefficient (SRCC, $\mathcal{S}$), Linear correlation coefficient (LCC, $\mathcal{L}$), and binary classification accuracy (ACC, $\mathcal{A}$). These metrics are essential for measuring regression performance and evaluating the model's capability in classifying aesthetic qualities as negative or positive.

**Benchmark Models.** We compare our method to 9 SOTA IAA models [1, 16, 17, 22, 41, 43, 48–50]. In selecting benchmarks, we prioritize generalization across multi-attributes and datasets rather than extreme optimization for a specific dataset; hence, we retrain 8 previously mentioned methods [1, 16, 17, 22, 41, 43, 48, 50] on multi-attribute datasets. Given that both our team and our peers in the field have faced the common bugs in reproducing the results of VILA[51], its exclusion from our analysis is regrettably unavoidable.

| Method | | Color Harmony | DoF | Light | Motion Blur | RoT | Vivid Color | Composition | Color | DoF | Light | Brightness | Colorful | Contrast | Noisiness | Sharpness |
|---|---|---|---|---|---|---|---|---|---|---|---|---|---|---|---|---|
| | | | | Attributes in AADB | | | | Attributes in PARA | | | | Attributes in SPAQ | | | | |
| Kong et al. | $\mathcal{S}\uparrow$ | 0.47 | 0.48 | 0.44 | 0.10 | **0.23** | 0.65 | 0.67 | 0.67 | 0.71 | 0.69 | 0.73 | 0.73 | 0.75 | 0.76 | 0.86 |
| | $\mathcal{L}\uparrow$ | 0.48 | 0.46 | 0.42 | 0.10 | 0.21 | 0.64 | 0.66 | 0.69 | 0.69 | 0.68 | 0.71 | 0.72 | 0.74 | 0.79 | 0.85 |
| | $\mathcal{A}\uparrow$ | 0.45 | 0.47 | 0.41 | 0.12 | 0.22 | 0.63 | 0.64 | 0.69 | 0.69 | 0.67 | 0.68 | 0.71 | 0.78 | 0.79 | 0.83 |
| $MP_{ada}$ | $\mathcal{S}\uparrow$ | 0.48 | 0.50 | 0.40 | 0.13 | 0.18 | 0.68 | 0.67 | 0.66 | 0.67 | 0.72 | 0.71 | 0.72 | 0.76 | 0.77 | 0.84 |
| | $\mathcal{L}\uparrow$ | 0.48 | 0.50 | 0.36 | 0.16 | 0.17 | 0.64 | 0.65 | 0.67 | 0.71 | 0.72 | 0.71 | 0.75 | 0.77 | 0.82 | 0.82 |
| | $\mathcal{A}\uparrow$ | 0.45 | 0.57 | 0.36 | 0.03 | 0.21 | 0.62 | 0.68 | 0.64 | 0.67 | 0.73 | 0.69 | 0.75 | 0.78 | 0.75 | 0.82 |
| Malu et al. | $\mathcal{S}\uparrow$ | **0.48** | **0.50** | 0.48 | 0.14 | 0.22 | 0.64 | **0.77** | **0.80** | 0.80 | 0.80 | **0.75** | 0.76 | 0.76 | 0.83 | **0.88** |
| | $\mathcal{L}\uparrow$ | 0.50 | 0.47 | 0.48 | 0.12 | 0.21 | 0.62 | 0.77 | 0.77 | 0.82 | 0.78 | 0.73 | 0.74 | 0.79 | 0.80 | 0.87 |
| | $\mathcal{A}\uparrow$ | 0.49 | 0.53 | 0.50 | 0.17 | 0.18 | 0.65 | 0.68 | 0.76 | 0.77 | 0.77 | 0.71 | 0.78 | 0.78 | 0.80 | 0.89 |
| NIMA | $\mathcal{S}\uparrow$ | 0.46 | 0.29 | 0.35 | 0.12 | 0.12 | 0.60 | 0.74 | 0.75 | 0.74 | 0.74 | 0.75 | 0.76 | **0.79** | 0.79 | 0.84 |
| | $\mathcal{L}\uparrow$ | 0.48 | 0.55 | 0.39 | 0.12 | 0.14 | 0.62 | 0.72 | 0.76 | 0.74 | 0.77 | 0.73 | 0.76 | 0.76 | 0.77 | 0.84 |
| | $\mathcal{A}\uparrow$ | 0.43 | 0.28 | 0.34 | 0.08 | 0.11 | 0.63 | 0.69 | 0.76 | 0.73 | 0.72 | 0.75 | 0.74 | 0.79 | 0.81 | 0.87 |
| MUSIQ | $\mathcal{S}\uparrow$ | 0.44 | 0.22 | 0.33 | 0.04 | 0.07 | 0.60 | 0.76 | 0.78 | **0.80** | 0.77 | 0.67 | 0.76 | 0.68 | **0.83** | 0.82 |
| | $\mathcal{L}\uparrow$ | 0.43 | 0.27 | 0.32 | 0.03 | 0.06 | 0.61 | 0.77 | 0.77 | 0.79 | 0.76 | 0.69 | 0.77 | 0.69 | 0.81 | 0.80 |
| | $\mathcal{A}\uparrow$ | 0.39 | 0.22 | 0.36 | 0.16 | 0.19 | 0.62 | 0.75 | 0.73 | 0.77 | 0.74 | 0.68 | 0.78 | 0.71 | 0.83 | 0.82 |
| TANet | $\mathcal{S}\uparrow$ | 0.48 | 0.48 | 0.48 | 0.14 | 0.22 | 0.63 | 0.74 | 0.77 | 0.78 | 0.76 | 0.73 | 0.75 | 0.79 | 0.80 | 0.85 |
| | $\mathcal{L}\uparrow$ | 0.47 | 0.47 | 0.48 | 0.17 | 0.18 | 0.68 | 0.74 | 0.79 | 0.77 | 0.76 | 0.74 | 0.76 | 0.82 | 0.79 | 0.83 |
| | $\mathcal{A}\uparrow$ | 0.45 | 0.49 | 0.44 | 0.17 | 0.18 | 0.64 | 0.72 | 0.75 | 0.79 | 0.75 | 0.72 | 0.75 | 0.78 | 0.78 | 0.85 |
| MaxViT | $\mathcal{S}\uparrow$ | 0.46 | 0.47 | 0.43 | 0.17 | 0.22 | 0.67 | 0.73 | 0.74 | 0.77 | 0.74 | 0.70 | 0.75 | 0.75 | 0.75 | 0.81 |
| | $\mathcal{L}\uparrow$ | 0.48 | 0.48 | 0.45 | 0.15 | 0.19 | 0.68 | 0.73 | 0.75 | 0.78 | 0.73 | 0.68 | 0.77 | 0.78 | 0.76 | 0.84 |
| | $\mathcal{A}\uparrow$ | 0.46 | 0.50 | 0.48 | 0.14 | 0.19 | 0.65 | 0.74 | 0.78 | 0.80 | 0.80 | 0.71 | 0.80 | 0.77 | 0.79 | 0.87 |
| EAT | $\mathcal{S}\uparrow$ | 0.46 | 0.49 | 0.49 | 0.15 | 0.22 | 0.65 | 0.74 | 0.76 | 0.76 | 0.77 | 0.71 | 0.75 | 0.74 | 0.77 | 0.84 |
| | $\mathcal{L}\uparrow$ | 0.46 | 0.51 | 0.47 | 0.15 | 0.20 | 0.65 | 0.75 | 0.76 | 0.75 | 0.75 | 0.74 | 0.78 | 0.78 | 0.79 | 0.86 |
| | $\mathcal{A}\uparrow$ | 0.47 | 0.53 | 0.46 | 0.10 | 0.20 | 0.68 | 0.77 | 0.72 | 0.79 | 0.75 | 0.74 | 0.73 | 0.78 | 0.77 | 0.83 |
| Ours | $\mathcal{S}\uparrow$ | **0.50** | **0.65** | **0.51** | **0.20** | **0.23** | **0.70** | **0.78** | **0.81** | **0.82** | **0.81** | **0.79** | **0.78** | **0.81** | **0.84** | **0.89** |
| | $\mathcal{L}\uparrow$ | 0.51 | 0.64 | 0.48 | 0.19 | 0.25 | 0.67 | 0.83 | 0.84 | 0.86 | 0.85 | 0.77 | 0.78 | 0.79 | 0.85 | 0.89 |
| | $\mathcal{A}\uparrow$ | 0.46 | 0.61 | 0.54 | 0.21 | 0.18 | 0.65 | 0.80 | 0.79 | 0.82 | 0.80 | 0.79 | 0.80 | 0.78 | 0.85 | 0.91 |
| Ours (SSL) | $\mathcal{S}\uparrow$ | 0.47 | 0.50 | **0.49** | **0.20** | 0.18 | **0.68** | 0.74 | 0.78 | 0.76 | **0.81** | 0.76 | **0.76** | 0.72 | 0.81 | 0.87 |
| | $\mathcal{L}\uparrow$ | 0.45 | 0.50 | 0.51 | 0.18 | 0.21 | 0.65 | 0.75 | 0.79 | 0.77 | 0.79 | 0.76 | 0.73 | 0.75 | 0.84 | 0.86 |

**Table 1: Comparison of the assessment effects of labeled atributes on the AADB, PARA and SPAQ datasets, where "Ours (SSL)" is assessed solely on self-supervision after training, with the red and blue bold numbers indicating the best and second best results.**

Furthermore, the absence of available code for TAVAR[49] necessitates our reliance on the metrics reported by TAVAR for comparative analysis with our method, as detailed in Table 2.

**Implementation Details.** The patch size is set to 32 during the training phase. The dimension $D$ for the decoder input tokens is fixed at 384, which is consistent across all embeddings. We use SGD with a momentum of 0.9 and initiate the learning rate at $10^{-4}$.

## 5.2 Performance Comparison

**Attribute Assessment on Labelled Data.** Table 1 presents the results on the three multi-attribute datasets. Our method outperforms other models for all attributes, surpassing the second-best method by an average of 4.48% in terms of SRCC.

**Attribute Assessment on Unlabelled Data.** To our knowledge, *we introduce the first method capable of label-free evaluation of aesthetic attributes through comparison.* To evaluate the performance, we train MAINet on three multi-attribute datasets with self-supervised learning (SSL), as detailed in Sec. 4.3.5, whereas labeled information is exclusively used for testing. As illustrated in Table 1, despite being trained on unlabeled data, our method achieves equal or superior performance compared to other methods trained on labeled data, thereby confirming its effectiveness.

**Cross-dataset Evaluation.** To evaluate the generalization ability of MAINet, the cross-dataset evaluations are conducted on four datasets. Table 2 demonstrates that MAINet outperforms on most datasets, especially when trained on a small dataset (AADB) with limited images.

## 5.3 Ablation Studies

Table 3 presents ablation experiments conducted on three multi-attribute IAA datasets: AADB, SPAQ and PARA.

**Effectiveness of Four-tier Training Strategy for RSM.** In Section 4.3.1, we outlined a sequential training strategy across four tiers, culminating in a gap between the predicted boundaries $B_l^1$ and $B_u^1$ that is approximately 1. By contrast, if we solely utilize the training data in one tier (while maintaining the same volume of data) to optimize the RSM, the final predictive performance declines by 4.9% in SRCC and 4.2% in PLCC. This observation confirms that the training strategy significantly enhances the model's ability to identify appropriate reference images.

**Importance of RSM.** The significance of the RSM module was validated by employing a simple random selection of reference images for comparison. These experiments reveal that our RSM significantly influences the assessment results. thereby indirectly supporting the 'Compromise I' hypothesis which suggests that image sequences with a shared theme exhibit transitivity akin to mathematical inequalities. As shown in Table 3, using images chosen by RSM resulted

| Test on | Method (pretrained on AADB) | | | | | | | | | |
| | Kong *et al.* | $MP_{ada}$ | Malu *et al.* | NIMA | MUSIQ | TANet | MaxViT | EAT | TAVAR | Ours |
| --- | --- | --- | --- | --- | --- | --- | --- | --- | --- | --- |
| AADB | 0.68 | 0.76 | 0.70 | 0.72 | 0.75 | 0.75 | 0.75 | 0.77 | 0.76 | **0.78** |
| SPAQ | 0.44 | 0.50 | 0.50 | 0.51 | 0.58 | 0.55 | 0.58 | 0.55 | N/A | **0.61** |
| PARA | 0.52 | 0.52 | 0.56 | 0.53 | 0.55 | **0.66** | 0.60 | 0.62 | N/A | 0.65 |

**Table 2: Cross-dataset evaluations of 9 methods on the AADB, PARA and SPAQ datasets . "N/A" signifies the absence of official code, preventing us from completing cross-dataset comparisons on this method.**

| Method | AADB | | SPAQ | | PARA | |
| | $\mathcal{S}\uparrow$ | $\mathcal{L}\uparrow$ | $\mathcal{S}\uparrow$ | $\mathcal{L}\uparrow$ | $\mathcal{S}\uparrow$ | $\mathcal{L}\uparrow$ |
| --- | --- | --- | --- | --- | --- | --- |
| w/o Four-tier Training | 0.74 | 0.75 | 0.88 | 0.88 | 0.87 | 0.88 |
| w/o RSM | 0.65 | 0.65 | 0.88 | 0.84 | 0.85 | 0.87 |
| w/o PIM | 0.74 | 0.74 | 0.90 | 0.91 | 0.89 | 0.89 |
| 2D PIM | 0.75 | 0.74 | 0.90 | 0.90 | 0.91 | 0.89 |
| w/o APM | 0.77 | 0.76 | 0.89 | 0.90 | 0.88 | 0.90 |
| half-intensity opeartion | 0.72 | 0.72 | 0.86 | 0.88 | 0.88 | 0.89 |
| Fully model | **0.78** | **0.79** | **0.92** | **0.91** | **0.92** | **0.92** |

**Table 3: Ablation studies conducted on AADB, PARA and SPAQ.**

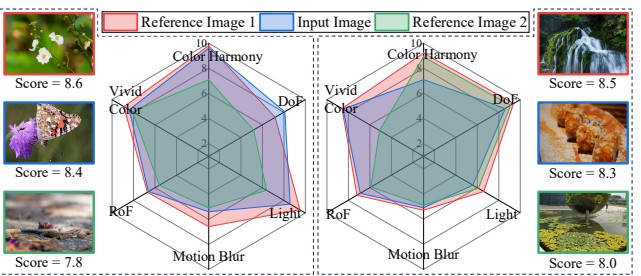

**Figure 4: Examples for attribute assessment show that higher aesthetic scores correspond to more coverage on the radar maps.**

in an 11.7% increase in SRCC and a 14.9% increase in LCC compared to randomly selected ones. This logical finding implies that randomly chosen reference images often have aesthetic disparities or fail to adequately position the input image, thus lacking valuable information for aesthetic comparisons.

**Effectiveness of PIM.** In addition, incorporating PIM enhances both SRCC and LCC by 1.8% and 3.4%, respectively. To further validate the reliance of our comparative framework on PIM, we also employed traditional 2D positional coding as an alternative to PIM, resulting in a decrease of 1.1% in SRCC and 2.3% in LCC. The unsatisfactory performance can be attributed to the model's inability to effectively capture the source information of image blocks, thereby causing confusion during the comparison process.

**Effectiveness of APM.** Finally, the incorporation of APM and augmented dataset results in a significant improvement of 3.1% and 4.5% in SRCC and LCC, respectively. This confirms that including attribute information enhances the model's accuracy in aesthetics assessment. Moreover, a decrease in the intensity of image editing operations yields a corresponding decline of 3.1% in SRCC and 2.7% in LCC, as illustrated in row 6 of Table 3.

## 5.4 Visual Analysis

**Predictions for Images.** The examples in Fig. 4 illustrate the multi-attribute IAA predictions. Similar to human cognition, MAINet assigns higher scores to images that exhibit superior performance across the majority of attributes.

**Saliency Maps.** The Grad-CAM is applied to visualize saliency maps in Fig. 5. MAINet can perceive multi-attribute features differentially, such as the DoF attribute which focuses on areas with salient objects and Vivid Color attribute which focuses on areas with vibrant colors.

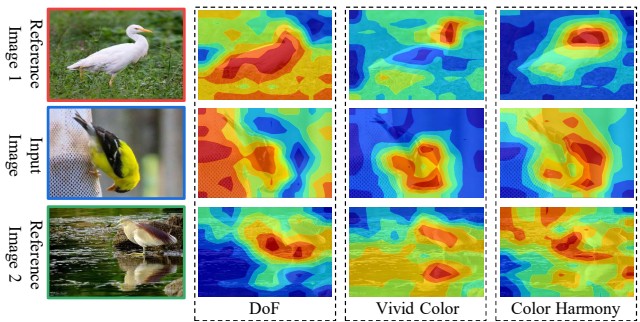

**Figure 5: Cases of attribute saliency maps demonstrate the varying perception of MAINet towards different attributes.**

## 6 CONCLUSION

This paper focuses on quantifying aesthetics and proposes a law called SR-IAA. To validate this law, we have developed a multi-attribute IAA framework that incorporates modules for automated reference image selection and integration of aesthetic attributes into the comparison process. To ensure reliable training, we have devised a novel scheme for constructing datasets and a progressive training strategy. Experimental results show that MAINet, guided by our proposed law, outperforms conventional state-of-the-art methods.

However, MAINet only serves as a preliminary verification method but entails inherent compromises due to the limitations in existing datasets. Additionally, the limited availability of data that fulfills the requirements of SR-IAA, inadequate experimental validation, and length constraints impede this paper from establishing a comprehensive groundwork for SR-IAA; therefore, we eagerly anticipate undertaking an in-depth exploration of this subject matter in another future publication.

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
