# OpenReview forum: ""Special Relativity" of Image Aesthetics Assessment: a Preliminary Empirical Perspective"
_acmmm.org/ACMMM/2024/Conference — MM2024 Poster_

### Official Review · Reviewer_QmKt · 2024-05-14

**Rating:** 5
**Confidence:** 4

**Summary:**

This study delves into the fundamental issue of image aesthetic assessment, specifically the challenge of quantifying image aesthetics.

The paper first address 3 key questions regarding whether aesthetic quality can be accurately quantified, the essence of quantifying aesthetics, and how aesthetics can be precisely measured, and then propose the “Special Relativity” of Image Aesthetics Assessment (SR-IAA). This principle encompasses the following three main points:
(1) Image aesthetic quality can be derived through the comparison of two images, i.e., through a referential assessment.
(2) Quantification of aesthetics can be categorized into no-reference aesthetics and reference-based aesthetics.
(3) No-reference aesthetics evaluation focuses on overall scores, while reference-based aesthetics assesses changes in multiple aesthetic attributes between two images.

Based on above mentioned, the paper establishes an algorithmic framework for multi-attribute aesthetic assessment, and achieving promising performance on conventional benchmarks.

For selecting reference images for aesthetic images to be evaluated, a Reference Selection Module (RSM) is constructed. This module selects reference upper and lower bound images with quality scores gradually approaching from data of the same theme. An Attribute Perception Module (APM) is built to predict various degradation operations and levels, as well as aesthetic-related attribute changes. The learning process is conducted through a self-supervised manner.

**Strengths:**

(1)	The paper attempts to address in more depth the important but understudied fundamental issues of IAA for the first time, conducting a thorough analysis and providing solutions.

(2)	The authors proposed an interesting framework that is novel and achieves a promising performance.

(3)	The experimental results are impressive, demonstrating that MAINet, guided by the proposed law, significantly outperforms conventional state-of-the-art methods of Multi-attribute IAA.

(4)	The paper is well-organized and easy to understand.

**Limitations:**

(1)	It would be beneficial to include additional descriptions of the entire assessment process of RSM and APM.

(2)	Given the constraints inherent in the current datasets, this paper employed specialized processing techniques on existing aesthetic datasets to align them with the theoretical framework outlined. However, It is advisable for the authors to elaborate on potential solutions and enhancements while addressing the current limitations.

(3)	Typos:

-In the section “Importance of RSM,” the “Compromise I” should be bolded to maintain a consistent style with the preceding text.

-There should be no space before the first paragraph of each section; please revise the section 3.

-Similar formatting glitches are present in the subsections, such as in section 4.3.2.

**Suitability:**

3

---

### Official Review · Reviewer_9DpU · 2024-05-21

**Rating:** 4
**Confidence:** 3

**Summary:**

The article presents a novel approach to image aesthetics assessment (IAA) termed the "Special Relativity" of IAA (SR-IAA), which addresses key questions about the quantification of aesthetic qualities. The authors introduce a Multi-Attribute IAA Framework (MAINet) that serves as an empirical validation of SR-IAA, achieving state-of-the-art performance on existing datasets. MAINet is designed to simulate two abilities of human perception: assessing aesthetics without reference and with reference, and it incorporates automated reference image selection and aesthetic attribute integration. The framework is trained using a progressive strategy, and experiments demonstrate its superiority over other IAA methods. The paper concludes by acknowledging the preliminary nature of the work and the limitations imposed by existing datasets, suggesting further exploration is needed.

**Strengths:**

This is an interesting work that conducts in-depth analysis of quantitative issues for IAA, and discusses and analyzes from three special perspectives: philosophy, psychology and mathematics.
1. **Innovative Framework**: The paper introduces a novel framework, MAINet, which is guided by the proposed "Special Relativity" of Image Aesthetics Assessment (SR-IAA). This framework aims to address the fundamental questions of quantifying aesthetics and offers a new perspective on image aesthetics assessment.
2. **State-of-the-Art Performance**: The authors claim that their framework achieves state-of-the-art performance on existing datasets, outperforming the second-best performance by significant margins in terms of Spearman Rank Correlation Coefficient (SRCC). This suggests that their approach is highly effective and pushes the boundaries of current research in this field.
3. **Comprehensive Analysis**: The paper provides a comprehensive analysis of the existing methodologies and their limitations in the context of image aesthetics assessment. It systematically addresses three fundamental questions related to aesthetic quantification and provides a structured response to each, demonstrating a deep understanding of the subject matter.

These strengths highlight the paper's potential contribution to the field of image aesthetics assessment through its innovative approach, empirical validation, and in-depth analysis.

**Limitations:**

The article presents a study on image aesthetics assessment (IAA) and introduces a framework called MAINet. However, it acknowledges several limitations:
1. **Dataset Limitations**: The article mentions that existing datasets primarily rank images based on scores, which contradicts the principles of the "Special Relativity" of IAA (SR-IAA). The authors had to make compromises to align the dataset with their framework, which may affect the generalizability of the results.
2. **Scalability of Multi-Attributes**: The existing datasets have limited scalability when it comes to incorporating new or redefined multi-attributes. The authors propose a method to approximate the difference in aesthetic attributes between original and edited images, but this approach may not fully capture the complexity of aesthetic perception.
3. **Reliance on Average Ratings**: Most available datasets are annotated using average ratings from multiple assessors. The paper suggests that this approach can be approximated to represent a "smoothed" viewer, but it may introduce new challenges and might not accurately reflect individual differences in aesthetic perception.

**Suitability:**

3

---

### Official Review · Reviewer_v4wN · 2024-05-31

**Rating:** 2
**Confidence:** 3

**Summary:**

This paper proposes a law composed of three elements to address three fundamental questions in Image Aesthetics Assessment (IAA). Based on this law, it introduces an IAA model called MAINet. MAINet consists of five modules and takes triplets as input to determine aesthetic scores. Experimental results demonstrate that MAINet outperforms existing methods in attribute assessment.

**Strengths:**

The proposed method achieves superior performance in attribute assessment compared to existing methods.

**Limitations:**

This paper raises fundamental questions regarding IAA and proposes laws and an IAA model that could address these questions. However, it is difficult to say that the questions have been adequately answered, and it is also challenging to see that the proposed methods were developed based on these answers.

1. The author argues in the first question that quantifying aesthetics is highly subjective and that it is necessary to establish certainty. As a method for this, they propose the RSM to create Triplets. However, can this be considered a sufficient method for quantifying aesthetics? Since it only evaluates good and bad within images of the same class, it cannot provide a fundamental answer for quantifying aesthetics. Moreover, the author is simply using existing datasets, which already use ground truth scores determined by subjective criteria. Therefore, demonstrating good performance in the context of existing datasets cannot be considered a fundamental answer to the first question. A more convincing answer is needed for the first question, and high performance in cross-dataset evaluation is not a enough answer either.

2. In the second question, the author mentions the non-transitive nature of IAA, but the method used in the paper is still based on numerical scores. Since the paper proceeds by stating in Compromise I that 'a common theme exhibits a certain level of transitivity,' it cannot be considered an adequate answer to the second question.

3. SR-IAA II and III address cases without references, but MAINet appears to be a model proposed primarily for cases with references, based on Triplets. Where can we find experimental or practical details regarding cases without references?

4. What is the reason for training RSM? The data already has ground truth scores, so it seems like we could simply group the data within the same class according to these scores. Why is training involved?

5. In SR-IAA III, it is mentioned that two images are compared based on multiple aesthetic attributes, and it seems that APM aims to reflect these aspects. However, I believe the attributes mentioned in SR-IAA III should be more comprehensive than just noise or color effects in Figure 3. I think the elements in Figure 3 are closer to IQA than IAA.

6. What is the reason for training PIM? Even with Triplets, the order of the data is fixed, so it doesn't seem like there would be any mixing of patches, even if you used a common sinusoidal position embedding.

7. The reason for the superior performance of the proposed method in Table 1 seems to be that it separately learns multiple attributes through APM, while the other methods do not, leading to lower performance. Therefore, it does not seem to be a fair comparison.

8. Regarding IAA performance, the proposed method scores 0.78 on AADB in Table 2, while EAT scores 0.77, which does not appear significantly different from existing methods.

**Suitability:**

3

---

### Meta-Review · Area_Chair_rmRo · 2024-06-30

**Recommendation:** Accept (Poster)
**Confidence:** 3

**Metareview:**

The final ratings are 1 borderline reject (R1), 1 weak accept (R2), and 1 accept. The reviewers acknowledged that the proposed method is interesting and shows good performance. However, R1 raised a concern in the final justification that the SR-IAA law lacks sufficient evidence/justification and the proposed model is not directly related to the SR-IAA law. After carefully checking the comments, paper, and rebuttal, AC finds this concern is reasonable to some extent.